# Sex-Related Predisposition to Post-Traumatic Stress Disorder Development—The Role of Neuropeptides

**DOI:** 10.3390/ijerph19010314

**Published:** 2021-12-28

**Authors:** Małgorzata Lehner, Anna Skórzewska, Aleksandra Wisłowska-Stanek

**Affiliations:** 1Department of Neurochemistry, Institute of Psychiatry and Neurology, 9 Sobieskiego Street, 02-957 Warsaw, Poland; mlehner@ipin.edu.pl (M.L.); skorzews@ipin.edu.pl (A.S.); 2Centre for Preclinical Research and Technology (CEPT), Department of Experimental and Clinical Pharmacology, Medical University of Warsaw, 1B Banacha Street, 02-097 Warsaw, Poland

**Keywords:** PTSD, sex differences, CRF, orexin, oxytocin, neuropeptide Y

## Abstract

Post-traumatic stress disorder (PTSD) is characterized by re-experiencing a traumatic event, avoidance, negative alterations in cognitions and mood, hyperarousal, and severe functional impairment. Women have a two times higher risk of developing PTSD than men. The neurobiological basis for the sex-specific predisposition to PTSD might be related to differences in the functions of stress-responsive systems due to the interaction between gonadal hormones and stress peptides such as corticotropin-releasing factor (CRF), orexin, oxytocin, and neuropeptide Y. Additionally, in phases where estrogens levels are low, the risk of developing or exacerbating PTSD is higher. Most studies have revealed several essential sex differences in CRF function. They include genetic factors, e.g., the CRF promoter contains estrogen response elements. Importantly, sex-related differences are responsible for different predispositions to PTSD and diverse treatment responses. Fear extinction (the process responsible for the effectiveness of behavioral therapy for PTSD) in women during periods of high endogenous estradiol levels (the primary form of estrogens) is reportedly more effective than in periods of low endogenous estradiol. In this review, we present the roles of selected neuropeptides in the sex-related predisposition to PTSD development.

## 1. Introduction

Post-traumatic stress disorder (PTSD) is a mental health condition triggered by a traumatic event [1,2]. Symptoms of PTSD are re-experiencing trauma, avoidance of trauma-associated stimuli, general anxiety, emotional numbing, and hyperarousal [3]. The key feature of this disorder is impaired extinction (defined as a decrease in the frequency and amplitude of conditioned responses) and enhanced emotional memory consolidation during trauma. The pathophysiology of PTSD is associated with enhanced feedback sensitivity of the hypothalamic-pituitary-adrenal (HPA) axis, hypersecretion of corticotropin-releasing factor (CRF), subnormal cortisol levels (hypocortisolemia), and noradrenergic hyperactivity [1,4].

A critical sex-specific predisposition to the occurrence of PTSD has been identified. Women have a two times greater risk of developing PTSD and are more likely to suffer a more chronic course of the disorder [5,6,7,8,9]. Sex differences in PTSD prevalence are evident even when men and women experience the same type of trauma, such as accidents, terrorism, and natural disasters [7,9]. Moreover, epidemiological studies suggest that women may have a higher risk for developing PTSD or exacerbating their present symptoms during phases of their lives, when estrogen levels are low: puberty, menses, postpartum, and menopause [10]. In addition, women who experienced the initial trauma during the luteal phase (low estrogen levels) reported more significant difficulties with flashbacks [11]. Researchers have hypothesized that the higher prevalence of PTSD in women may be a consequence of estrogen-related differences in neuroendocrine and stress response systems [12]. Moreover, the functional magnetic resonance (fMRI) that indicates that stressful events would cause a greater increase in the arousal network (e.g., locus coeruleus (LC) and periaqueductal gray) of women than men following exposure to aversive (threat) stimuli probably due to different sensitivity of these structures depending on the estrogen concentration [13].

Considering that the stress response is variable and depends on the changes in hormones concentration, in this review, we focus on the role of selected hypothalamic hormones in the estrogen dependent predisposition, course, and treatment of PTSD in females. The major regulator of the HPA axis is a corticotropin-releasing factor, the pituitary hormone adrenocorticotropic hormone (ACTH), and the negative feedback effects of adrenal glucocorticoids, but other hypothalamic neuropeptides, including oxytocin, orexin, and neuropeptide Y (NPY) can affect HPA axis activity by influencing the expression and secretion of CRF, modulating pituitary function or adrenal steroidogenesis. Hypothalamic peptides and the HPA axis reciprocally control each other’s activities; moreover, gonadal hormones could affect their function. The interaction between sex hormones, the HPA axis, and hypothalamic hormones is complex and occurs on both genetic—by controlling the expression of peptides via interaction with gene promotors—and epigenetic levels. Patients with PTSD exhibit an altered balance between excitatory (CRF and orexins) and anti-stress hormones (oxytocin and NPY) [5,14,15]. In this review, we decided to introduce this poorly explored problem.

## 2. Hypothalamic Stress Neuropeptides

### 2.1. CRF

CRF is a neuropeptide released from the paraventricular nucleus of the hypothalamus (PVN) that activates the HPA axis and regulates stress-related behaviors. CRF stimulates the secretion of adrenocorticotropic hormone, which acts on the adrenal cortex to produce glucocorticoids (e.g., cortisol in humans and primates and corticosterone in rodent species) [16,17,18]. Via GR (glucocorticoid) and MR (mineralocorticoid) receptors, glucocorticoids regulate negative feedback to the HPA axis by inhibiting the release of CRF and ACTH. Moreover, CRF centrally modulates the activity of various brain regions (hippocampus, amygdala and locus coeruleus) that regulate behavioral responses to stress and cognitive function [19,20,21]. Increased cerebrospinal fluid CRF concentrations have been reported in patients with PTSD [16,17,18].

Most studies have revealed several essential sex differences in CRF function. Preclinical and clinical data suggest greater sensitivity of CRF neurons in regions of the hypothalamus and bed nucleus of the stria terminalis (BNST) in female rats than in male rats to environmental stress and corticosterone levels [21,22,23,24]. Different predispositions to stress-related psychiatric disorders might be partially attributed to interactions between gonadal hormones and stress hormones [25,26]. Estrogens enhance activation of the CRF gene in stress. Estradiol also regulates CRF binding protein (CRF-BP) mRNA expression and increases the excitability of CRF neurons in the PVN [27]. Evidence suggests that in animal models (primates and rats), the HPA axis of females is more sensitive than that of males to the effects of higher levels of CRF during phases of the estrous cycle characterized by high ovarian hormone levels. Moreover, gonadal hormones have an important role not only in adult individuals but also in the perinatal period, as estradiol and testosterone exposure organizes sex differences in the CRF system in PVN [28]. The structures containing CRF receptors are also characterized by the presence of estrogen receptors; thus, gonadal hormones are potentially able to regulate CRF receptors [29]. For example, estrogen treatment significantly increased CRF mRNA expression in the amygdala and the BNST of ovariectomized mice [30,31]. Estrogen increases the expression of CRF1 and CRF2 in breast cancer cells [32] and probably in the brain [33]. Many estrogen effects are mediated by estrogen receptors (ER) that bind to specific estrogen response elements (EREs) in target promoters [26,31]. The CRF promoter contains EREs; therefore, it is assumed that estrogen may directly affect CRF gene expression [34]. Several studies have documented decreased CRF mRNA expression in the PVN of ovariectomized animals and its restoration after estradiol replacement treatment, suggesting a stimulatory effect of estrogen on CRF [34,35,36,37].

CRF action is mediated by CRF1 and CRF2 receptors. Evidence for sex differences in CRF receptor expression, distribution, trafficking, and activation in brain regions that might be associated with different predispositions to PTSD has been reported [25,38]. In the cingulate cortex and amygdala, CRF1 receptor binding is higher in adult female rats. In comparison, CRF2 receptor binding is higher in the amygdala and the hypothalamus in male rats [39]. This distribution is probably related to an increase in anxiety-like behaviors in females under stress compared to males [5].

Studies have revealed that stress-induced increases in CRF levels contribute to neuropsychiatric disease development through the excess activation of its type 1 receptor [40]. Moreover, pharmacological antagonism of CRF1 suppresses the stress response and decreases anxiety- and depressive-like behaviors [41]. Importantly, pharmacological intervention might exert an anti-anxiety effect, but some internal homeostatic processes, such as receptor trafficking, potentially attenuate the stress response and anxiety. These processes are sex-specific and occur in the locus coeruleus. Data suggest that the higher predisposition of women to PTSD might result from stronger activation of the locus coeruleus by CRF than in men. LC neurons are protected from the effects of excessive CRF by internalizing CRF1 receptors [42]. Females have greater activation of LC circuits, which may be associated with the decreased internalization of CRF1 receptors under stress conditions compared to males [25,38].

### 2.2. Orexin

Orexins (ORX), which are also called hypocretins, are excitatory neuropeptides involved in the neuroendocrine response to stress, arousal, food intake, cognitive flexibility, sleep control, emotional memory, and reproduction [43,44]. Both stressful stimuli and CRF increase orexin neurons activity [45,46]. The orexin system is composed of endogenous neuropeptides, orexin A and B, and associated orexin type 1 and 2 receptors (OX1R and OX2R, respectively) [47]. Orexins are synthesized in the hypothalamus, while orexin receptors are widely distributed in the brain. OX1R receptors are expressed in the cingulate cortex, the hippocampus, BNST, and amygdala [48]. Orexins are presumed to be involved in cognitive functions via stimulation of pyramidal neurons in the prefrontal cortex and dentate gyrus of the hippocampus [48,49]. ORX A administration into the hippocampus reverses memory disruption under conditions of orexin deficiency [50]

### 2.3. Dysfunction of the Orexin System Has Been Reported in PTSD

A correlation study suggested that high ORX A levels in plasma were associated with spontaneous recovery of recent memory after extinction [51,52]. In particular, ORX expressed in the amygdala might be the target of fear memory and anxiety disorders [53]. During acute stress, orexin increases HPA activity and promotes the neuroendocrine response. In contrast, repeated stress involves more complex mechanisms contingent on the type, intensity, and duration of the stimuli, which indicate the inherent plasticity of the orexin system [54].

Preclinical and clinical data highlight sex differences in the orexin system, as female rats showed higher baseline levels of orexin precursor prepro-orexin mRNA, orexin neuron activation (measured as c-Fos immunoreactivity) in the hypothalamus, and orexin A levels in the cerebrospinal fluid than male rats [55]. Elevated orexin levels in female rats are responsible for the increased HPA responses to repeated stress and the stress-induced impairments in cognitive flexibility [55]. The literature suggests that gonadal hormones might be involved in the sex-specific interaction between the HPA axis and orexin [55]. Regulation of orexin expression in female rats depends on their reproductive status. Studies have indicated that female rats display higher prepro-orexin mRNA levels during proestrus (high level estrogen phase), but ovariectomized females showed no reduction in prepro-orexin mRNA expression [56]. Moreover, in women, plasma orexin levels increase significantly during menopause [57].

Studies have reported increase in OX1R and OX2R expression in the PVN of female rats [58]. Orexins regulate the stress response by increasing ACTH release from the pituitary through OX1R and OX2R receptors and stimulating glucocorticoid release via OX1R in the adrenal gland [59,60,61]. In the chronic unpredictable mild stress model, a significant increase in OX1R expression was observed in the frontal cortex of female rats, but not male rats [62]. In the PVN, glucocorticoid receptors act directly on the orexin promoter to increase prepro-orexin expression, providing a regulatory mechanism that exclusively controls orexin system activity in females [55].

## 3. Anti-Stress Hormones

### 3.1. Oxytocin

Oxytocin (OT) is a neuropeptide synthesized in the hypothalamus. In addition to the classical functions of oxytocin, such as inducing uterine contractions during labor and milk ejection during nursing, it regulates stress coping behaviors through its anxiolytic action, which is essential for maternal behavior and social bond formation [63,64]. Neuronal projections from the hypothalamus send oxytocin to the posterior pituitary gland. In humans, oxytocin affects the central and basolateral amygdala, hippocampus, cingulate cortex, olfactory nucleus, and brainstem, regions expressing oxytocin receptors. The oxytocin and HPA axis are mutually regulating systems. Oxytocin inhibits the stress-induced activity of the HPA axis by promoting the return of cortisol levels to normal after stress is experienced [65]. OT has also been shown to decrease the activity of limbic regions that project to hypothalamic and brainstem regions and thus attenuate the fear response [66,67,68].

Stressful experiences might alter the function of the hypothalamus by decreasing the synthesis and release of endogenous OT [69]. Human studies suggest that outpatients with PTSD, both male and female, have reduced plasma oxytocin levels compared to healthy controls [70]. A reduced level of OT indicates decreased resiliency to stress [71]. Clinical trials with small samples suggest therapeutic effects of oxytocin administration on patients with PTSD [72,73]. Frijling [72] found that intranasal administration of oxytocin is a promising early preventive intervention for PTSD in individuals with an increased risk of PTSD. Oxytocin promotes fear extinction favoring better emotional control and enhanced cognitive performance by reducing hypervigilance, avoidance, and anhedonia [72,73].

Preclinical evidence indicates that OT expression differs in various brain structures (e.g., forebrain and hypothalamus) between males and females [74,75]. Moreover, the oxytocin promoter is regulated by estrogen [76]. Oxytocin disturbances are particularly associated with PTSD psychopathology in postpartum women [14,77]. Estrogen increases OT receptor expression and the release of this hormone, whereas androgens inhibit the release of OT under stress [78,79,80]. Sex effects might involve the differences in the number and binding affinity of OT receptors in specific neural networks [74].

### 3.2. Neuropeptide Y

Neuropeptide Y (NPY) regulates food intake, energy homeostasis, circadian rhythm, and cognition and has anxiolytic properties. High NPY expression is observed in the hypothalamus, septum, nucleus accumbens, periaqueductal grey, and LC [15]. Research indicates that the stress-protective effects of NPY are mediated by modulating neurotransmission in the amygdala, hippocampus, and cerebral cortex. NPY attenuates the stress-induced increase in the expression of norepinephrine biosynthetic enzymes and CRF release [81,82]. NPY neurons innervates ORX containing cells in the lateral hypothalamus [83]. NPY deficiency may upregulate ORX, activate noradrenergic neurons, and evoke heightened arousal [84]. Lower levels of NPY in the plasma and cerebrospinal fluid are associated with PTSD [85]. In preclinical studies, anxiolytic effects and improved fear extinction were documented after an intracerebroventricular (ICV) administration of NPY [86].

According to previous studies, NPY concentrations in the brain exhibit a sex-dependent pattern, with lower levels detected in female rats than in male rats under basal and stressed conditions [15]. In females, lower levels of the NPY peptide and lower expression of its receptor Y1R were observed in the hypothalamus, striatum, hippocampus, and plasma than in male rats. The lower expression of NPY in the hypothalamus might contribute to the increased activation of the HPA axis, resulting in greater susceptibility to stress-related disorders in females [15]. Additionally, the literature suggests a sex-specific difference in the expression of NPY receptor. Females have lower Y1R expression but higher Y1R-NPY binding affinity in the cerebral cortex than males [87]. Nahvi et al. [88] found that Y1R and CRFR1 mRNA levels increased in the LC of female rats compared with unstressed control rats 1 week or more after exposure to a single prolonged stress (PTSD model). In females, Y1R receptors appear to attenuate the effects of CRF1 receptors. Furthermore, no changes in Y1R expression were found in males, suggesting the potential importance of these receptors in females [15,82]. Estrogen contributes essentially to the regulation of Y1R expression because its gene contains of estrogen response elements. Consequently, during the high-estrogen phase (proestrus), an increase in Y1R expression was detected in the hypothalamus compared to other estrous cycle phases [15,89,90]. Furthermore, the female to male ratio of PTSD prevalence was highest among adolescents and young adults approximately (3:1) and decreased (2:1) among older adults (66 to 70 years old) [91,92]. 

In the Table 1 we presented the influence of estrogens and HPA axis on the expression of selected hypothalamic peptides. In the Figure 1 there are presented the differences in the neuropeptides and receptors in brain structures of women or men with PTSD. 

Table 1 shows influence of estrogens and HPA axis on the expression of selected hypothalamic peptides, Table 2 shows interaction between hypothalamic peptides. Figure 1 shows changes in neuropeptides’ activity and their receptors in the brain. 

## 4. Epigenetic Changes

Epigenetics is a sequence-independent heritable DNA change that may be triggered by environmental factors [100]. Epigenetic mechanisms include DNA methylation, histone modifications, nucleosome repositioning, higher-order chromatin remodeling, noncoding RNAs (microRNAs), and RNA and DNA editing [101]. Some epigenetic modifications occur in a sex-specific matter and might be associated with a predisposition to PTSD [102]. Preclinical studies showed that in the model of chronic mild stress, stressed females had a higher level of total DNA methylation across the CRF gene in the PVN than stressed males [102]. Moreover, some epigenetic changes are potential biomarkers of disease or are valuable for monitoring treatment, e.g., the response of individuals with PTSD to treatment is associated with decreases in the methylation of FKBP5 (FK506-binding protein 5, the protein that binds to and inhibits glucocorticoids) [4]. Interestingly, studies of the offspring of Holocaust survivors indicate that maternal PTSD has a more significant contribution than paternal PTSD to an increased risk of PTSD, probably due to epigenetic mechanisms [103].

MicroRNAs, critical regulators of gene expression involved in neuronal development, synapse formation, and synaptic plasticity, have also been linked to the regulation of neurobiological systems underlying anxiety processing in PTSD [104,105]. MicroRNAs are a class of small noncoding RNAs that mediate cleavage or translational repression of target mRNAs [106]. The binding of microRNAs decreases the stability of target mRNAs or inhibits their translation to downregulate mRNA expression [107]. A potential role for miR-19b in regulating genes associated with delayed and exaggerated fear was reported [108]. Previous data showed that miR-19b was under the transcriptional control of the estradiol [109]. Male animals exposed to single prolonged stress exhibited higher levels of circulating miR-19b than control male animals. In turn, female animals exposed to unpredictable sound stress displayed lower levels of miR-19b compared with control female animals [109]. Translational research approaches identified miR-19b as a potential marker for PTSD that is differentially expressed in males and females after trauma or stress. In a clinical study, the miR-144-5p expression levels in the blood plasma of patients suffering from depression/anxiety were lower than those in healthy controls [110]. Interestingly, miR-144-5p increased to levels observed in healthy controls following eight weeks of psychotherapy, suggesting that miR-144-5p might be a potential biomarker of treatment success [105]. Additionally, miR-138-5p was overexpressed in subjects with PTSD compared to controls, and miR-1246 was downregulated in patients with PTSD compared to resilient subjects. These findings suggest that these microRNAs might be potentially relevant to diagnosing PTSD [111].

## 5. PTSD Therapy in Women

According to the data, less than 50% of individuals with PTSD remit spontaneously [4]. In the treatment of PTSD, the most important and well-established role has been identified for psychological therapies with (or without) pharmacological support. These treatments reduce symptoms severity [4]. The known predictor of exposure therapy is fear extinction. This process depends on the acquisition of new inhibitory learning, where fear memory is weakened by the successive presentation of a fear-eliciting conditioned stimulus without an aversive stimulus [112]. Fear extinction involves the weakening (long-term depression) of previously potentiated synapses [113]. Fear processes and fear extinction are strongly regulated by circulating sex hormones; however, the specific effects of sex hormones on fear extinction remain poorly understood [114].

Both preclinical and clinical studies suggest that fear extinction depends on the estrogen levels [115,116]. Neuroimaging studies show greater activation of neural networks involved in fear induction during the early follicular phase of the menstrual cycle (low estrogen levels) compared to mid-cycle (high estrogen levels) [117]. However, female rats subjected to conditioning in the proestrus and estrus stages (high estrogen levels) extinguished fear more rapidly than male rats and female rats in the diestrus stage (low estrogen levels) [118]. Additionally, a high level of estrogen protected against the extinction deficit observed in women with PTSD [119]. Studies in both humans and animals support this observation by showing that low estrogen levels are associated with deficits in fear extinction recall compared with phases characterized by higher estrogen level. Psychotherapy based on fear extinction is more effective in women during periods with high endogenous estradiol levels than in periods with low endogenous estradiol levels or the administration of hormonal contraceptives [120,121]. Thus, an analysis of exposure therapy in women using hormonal contraception seems very important. This behavioral effect might be associated with the effects of estrogens on the transcription of peptides that control the function of the HPA axis and play an important role in fear extinction, such as CRF and NPY.

In patients with more severe cases of PTSD, when psychotherapy alone is insufficient to achieve remission, selective serotonin reuptake inhibitors (SSRIs) are used. Only two medications are currently approved by the US Food and Drug Administration (FDA) for treating PTSD: paroxetine and sertraline [121]. SSRIs increase the expression of NPY in the brain, and this mechanism is related to fear extinction [122]. Hormonal fluctuations may modulate the activity of neuronal circuits relevant for fear extinction. Interestingly, a recent clinical trial that tested the effects of intranasal NPY on patients with PTSD found sex differences in response to this treatment. Women require higher doses of NPY than men to achieve a therapeutic effect [123].

Sleep disturbances are one of the most commonly reported and refractory symptoms of PTSD that might hamper the treatment of this disorder [124]. Studies focusing on trauma-related sleep disturbances in women are limited. They have shown therapeutic effects of cognitive-behavioral therapy for insomnia, imagery rehearsal therapy, or combinations of these techniques on sleep quality, insomnia, and nightmare severity in trauma-exposed patients [125]. Limited evidence also suggests the positive effects of drug-assisted psychotherapy with D-cycloserine, a partial agonist of the N-methyl-D-aspartate receptor, by enhancing extinction learning and attenuating insomnia severity [126].

## 6. Conclusions

PTSD is a mental disorder associated with the impaired extinction of traumatic episodes. PTSD occurs twice more frequently in women than in men. Additionally, epidemiological studies suggest that women may have a higher risk of developing or exacerbating PTSD during the phases of their lives when estrogen levels are low, such as puberty, menses, postpartum, and menopause [10]. The neurobiological basis of sex-related susceptibility to PTSD is associated with differences in the functioning of the HPA axis and hypothalamic neuropeptides that are partially dependent on estrogens. These sex differences in CRF system activity are partially attributed to circulating ovarian hormone levels because the CRF promotor contains estrogen response elements. Moreover, males and females exhibit differences in the expression of CRF1 and CRF2 receptors in the LC and limbic structures. CRF also regulates the actions of other neuropeptides that control emotional memories, such as orexin. Dysfunction of the orexin system has been reported in individuals with PTSD. In turn, the oxytocin and NPY expression is reduced in individuals with PTSD. These changes are associated with decreased resiliency in response to stress. Moreover, NPY has been reported to attenuate stress induced increases in the expression of norepinephrine biosynthetic enzymes and the activity of the HPA axis, counteracting the anxiogenic effects of CRF [81,82]. Estrogens not only influence the predisposition to PTSD but also affect its treatment. Fear extinction-based psychotherapy was reported to be more effective in women during periods of high endogenous estradiol levels than in periods with low endogenous estradiol levels or after the administration of hormonal contraceptives. This behavioral effect might be associated with the effect of estrogens on the transcription of peptides that control the function of the HPA axis and play an important role in fear extinction, such as CRF and NPY. Additionally, antidepressants that increase the effectiveness of psychotherapies, such as SSRIs alter the expression of hypothalamic neuropeptides, e.g., NPY.

## Figures and Tables

**Figure 1 ijerph-19-00314-f001:**
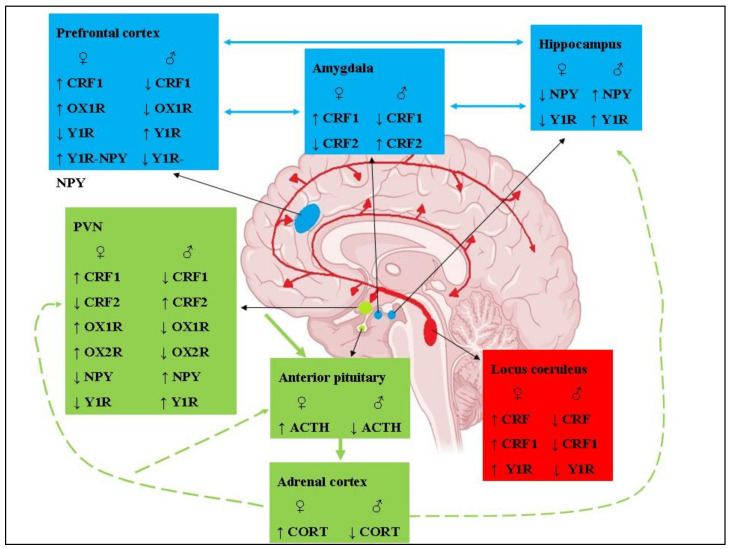
Changes in the activities of neuropeptides and their receptors in brain structures regulating cognition and the stress response that are responsible for the development of PTSD in women compared to men. The corticolimbic circuitry that mediates emotional responses to negative stimuli is shown in blue. The locus coeruleus norepinephrine system that initiates arousal responses to stress is shown in red. The HPA axis that regulates the neuroendocrine responses to stress is shown in green. Negative feedback (marked with green lines) is reduced in females and may further increase the release of glucocorticoids. 
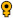
—females; 
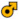
—males; ACTH—adrenocorticotropic hormone; CORT—cortisol; CRF- corticotropin-releasing hormone; CRF1, CRF2—CRF receptors type 1 and 2; OX1R, OX2R—orexin receptors type 1 and 2; NPY—neuropeptide Y; Y1R—NPY receptor type 1; Y1R-NPY—binding of NPY to the Y1R receptor, ↑—increase; ↓—decrease. The figure was adapted from Matchett et al. [95]. The permission has been obtained and there is no copyright issue.

**Table 1 ijerph-19-00314-t001:** Influence of estrogens and HPA axis on the expression of selected hypothalamic peptides.

Hypothalamic Peptide	Estrogen Influence	HPA Axis Influence
**CRF**	**Increases expression** of CRF via interaction with ERE [45]	Glucocorticosteroids via GR receptors inhibit CRF expression [93]
**CRF1 and CRF2 receptors**	**Increases** expression [33]	High CRF level **decreases expression of CRF receptors (down-regulation)** [94]
**Prepro-orexin**	No data	**Increases** expression by GR [55]
**Oxytocin**	**Increases** expression through action on oxytocin gene promoter [76]	No data
**Neuropeptide Y**	**Decreases expression via estrogen receptors** [92]	No data
**Neupopeptide Y receptors Y1R**	**Increases expression via interaction with ERE** [89]	No data

HPA—hypothalamic-pituitary-adrenal; ERE—estrogen response element; GR—glucocorticoid receptors, CRF—corticotropin releasing factor, CRF1, CRF2—CRF receptors type 1 and 2.

**Table 2 ijerph-19-00314-t002:** Interaction between hypothalamic peptides.

CRF	Orexin	Oxytocin	Neuropeptide Y
Activates orexin neurons in hypothalamus [46]	**Increases ACTH release in pituitary.** **Increases glucocorticoid in adrenal gland** [59,60,61]	Inhibits HPA axis [65]	Contradictory data decreases CRF release [81], activates CRF [96]
Inhibits oxytocin release [69]	Inhibits oxytocin neurons [97]	No data	Inhibits orexin neurons [83]
No data	Activates NPY neurons in the hypothalamus [98]	No data	Increase of oxytocin release [99]

CRF—corticotropin-releasing factor, ACTH—adrenocorticotropic hormone, HPA—hypothalamic-pituitary-adrenal, NPY—neuropeptide Y.

## Data Availability

No new data were created or analyzed in this study.

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
