# Peer review of "Sex-Related Predisposition to Post-Traumatic Stress Disorder Development—The Role of Neuropeptides"

_ijerph, 2021, doi:10.3390/ijerph19010314_

Round 1

Reviewer 1 Report

This is a narrative review that treats the role of neuropeptides in the gender-related predisposition to PTSD development. The topic is interesting and warrants future research. The authors describe the differences between genders in the stress related peptides CRF, orexin, oxytocin and neuropeptide Y as well as the epigenetic modifications in this matter. These observations could explain the higher incidence of PTSD in women and the influence of estrogen levels on PTSD predisposition, exacerbations and treatment results. The paper is well documented, information is clearly presented and it warrants publication.

There is one minor issue that needs to be clarified before publication:

  • Is Scheme 1 the same as Figure 1? If so, the same name should appear throughout the article.

Author Response

Referee 1

This is a narrative review that treats the role of neuropeptides in the gender-related predisposition to PTSD development. The topic is interesting and warrants future research. The authors describe the differences between genders in the stress related peptides CRF, orexin, oxytocin and neuropeptide Y as well as the epigenetic modifications in this matter. These observations could explain the higher incidence of PTSD in women and the influence of estrogen levels on PTSD predisposition, exacerbations and treatment results. The paper is well documented, information is clearly presented and it warrants publication.

There is one minor issue that needs to be clarified before publication:

  • Is Scheme 1 the same as Figure 1? If so, the same name should appear throughout the article.

Thank you for your valuable comments to the Manuscript. Scheme 1 is the same as Figure 1. We have corrected this mistake according to your suggestion.

Reviewer 2 Report

The English writing should be extensively edited. In specific, the word choice and grammar need to be improved.  Polish language service from native English speakers is suggested. It is also suggested that the author outline each paragraph and try to organize the logic flow better. More conjunction words are recommended to improve cohesion and coherence. Also, some misplaced modifiers are needed to be corrected. 

Author Response

Referee 2

The English writing should be extensively edited. In specific, the word choice and grammar need to be improved.  Polish language service from native English speakers is suggested. It is also suggested that the author outline each paragraph and try to organize the logic flow better. More conjunction words are recommended to improve cohesion and coherence. Also, some misplaced modifiers are needed to be corrected. 

Thank you very much for your valuable comments. As you suggested, we have sent the Manuscript to the professional editing service. The editing certificate is available in the attachment. As you suggested, we have outlined each paragraph and are trying to organize the logical flow better and improve the cohesion.

Reviewer 3 Report

The importance of the HPA axis, hormones, and neuropeptides are crucial for this review. However, the relationship of the above elements has not been shown. Authors should add this description or illustrate the HPA axis and its molecular relationship with estrogens and neuropeptides. 

1. The effect of neuropeptides is described chaotically. The presented thesis about sex-related predisposition to post-traumatic stress disorder development can only be proved if each element clearly relates to regulation by means of hormone signaling. This aspect is not sufficiently emphasized in the manuscript.

2. The work requires editorial corrections. There are numerous typos and punctuation mistakes in the text.

3. The figure is of poor quality, the idea behind this figure is also not clear. Please consider marking the differences between sex with different colors or in two diagrams. The caption "Figure 1" should be removed.

Author Response

Referee 3

The importance of the HPA axis, hormones, and neuropeptides are crucial for this review. However, the relationship of the above elements has not been shown. Authors should add this description or illustrate the HPA axis and its molecular relationship with estrogens and neuropeptides. 

We thank the reviewer for helpful and insightful comments. We have added two Tables to illustrate better the relation between estrogens, HPA axis elements, and hypothalamic peptides. In the text are also included more information how estrogens influence the CRF and oxytocin expression.

  1. The effect of neuropeptides is described chaotically. The presented thesis about sex-related predisposition to post-traumatic stress disorder development can only be proved if each element clearly relates to regulation by means of hormone signaling. This aspect is not sufficiently emphasized in the manuscript.

Thank you for your valuable remark. We try to make the flow of the Manuscript more logical. Moreover, in the current version of the Manuscript, we try to emphasize the relation between estrogens, the HPA axis, and hypothalamic hormones. We added the information about the influence of estrogens on CRF, oxytocin, and NPY and their receptor expression via action on gene regulation or trafficking of receptors. Based on literature data, we were not able to find any direct influence of estrogens on orexin gene expression. On the other hand, there is data that suggests the relationship between the estrogens phase and orexin expression (Grafe, L.A.; Cornfeld, A.; Luz, S.; Valentino, R.; Bhatnagar, S. Orexins Mediate Sex Differences in the Stress Response and in Cognitive Flexibility. Biol. Psychiatry. 2017, 81, 683-692). To better illustrate the influence of estrogens and HPA axis components on the expression of the hypothalamic peptides Table 1 was added. To complete the subject of the interaction between hypothalamic peptides, we supplemented the Manuscript with Table 2.

  1. The work requires editorial corrections. There are numerous typos and punctuation mistakes in the text.

We have sent the Manuscript to the American Journal Experts – the editing office. They have corrected style, typos, and punctuation mistakes. We enclose the certificate from AJE.

  1. The figure is of poor quality, the idea behind this figure is also not clear. Please consider marking the differences between sex with different colors or in two diagrams. The caption "Figure 1" should be removed.

We thank the reviewer for helpful and insightful comment. The Figure is corrected according to Referee suggestion. We have added the information on both males and females.

Round 2

Reviewer 2 Report

The Authors have very nicely answered and incorporated suggestions from the reviewers.

There are a few minor errors that need to be corrected before publication.

1: Page 1, last sentence, .....with flashbacks [11Researches have...... It should be [11]. Researches have....

2: Page 2, in 2.1 CRF section,  .....between gonadal hormones and stress hormones [26].27]. ..... It should be [26,27].

3: The first line indent should be consistent.

4: In page 4, why there is no 3.2 after oxytocin in the section of anti-stress hormones ?

5: After the 3. Anti-stress hormones, there is no 4. before directly jump to 5. Neuropeptide Y. 

6: In page 5, the format of table 1 and its associated ledgend should be corrected.

7. In page 6, the MicroRNAs should not be listed under Epigenetic changes.

Author Response

Referee 2

The Authors have very nicely answered and incorporated suggestions from the reviewers.

There are a few minor errors that need to be corrected before publication.

1: Page 1, last sentence, .....with flashbacks [11Researches have...... It should be [11]. Researches have....

2: Page 2, in 2.1 CRF section,  .....between gonadal hormones and stress hormones [26].27]. ..... It should be [26,27].

3: The first line indent should be consistent.

4: In page 4, why there is no 3.2 after oxytocin in the section of anti-stress hormones ?

5: After the 3. Anti-stress hormones, there is no 4. before directly jump to 5. Neuropeptide Y. 

6: In page 5, the format of table 1 and its associated ledgend should be corrected.

  1. In page 6, the MicroRNAs should not be listed under Epigenetic changes.

We thank the Reviewer for helpful and insightful comments. We are very sorry for so many editing errors. All errors that you kindly indicated in points 1-6 of the review were corrected.

Referring to point 7: Although this may raise doubts about microRNA a small non-coding RNA is an epigenetic modification, similarly to DNA methylation, histone modifications, nucleosome repositioning, higher-order chromatin remodeling (Gray, J.D.; Kogan, J.F.; Marrocco, J.; McEwen, B.S. Genomic and epigenomic mechanisms of glucocorticoids in the brain. Nat Rev Endocrinol 2017, 13, 661-673; Zhang, L.; Lu, Q.; Chang, C. Epigenetics in health and disease. Adv. Exp. Med. Biol. 2020, 1253, 3–55). Based on this information we decided to list it in the 4th chapter Epigenetic changes.

Reviewer 3 Report

The work is still being edited carelessly.

  1. No figure, the signature remains.
  2. Many linguistic mistakes. I don't feel qualified to judge about the English language and style, but, I notice only in the introduction:
  • A critical sex-specific predisposition to the occurrence of PTSD have been identified-> has been
  • Hypothalamic peptides and the HPA axis reciprocally control each other activities, moreover gonadal hormones could affect theirfunction. -> Hypothalamic peptides and the HPA axis reciprocally control each other activities, moreover, gonadal hormones could affect their function.
  • The interaction between sex hormones, the HPA axis and hypothalamic hormones….-> The interaction between sex hormones, the HPA axis, and hypothalamic hormones is complex and occurs on both genetic by controlling the expression of peptides via interaction with promo-tors of those peptides and epigenetic levels.
  • The interaction between sex hormones, the HPA axis and hypothalamic hormones is complex and occur on at both genetic by controlling the expression of peptides via interaction with promo-tors of those peptides and epigenetic levels -> The interaction between sex hormones, the HPA axis, and hypothalamic hormones are complex and occurs on both genetic by controlling the expression of peptides via interaction with promo-tors of those peptides and epigenetic levels.

Moreover:

„This hypothesis is endorsed by a meta-analysis of functional magnetic resonance (fMRI) data suggesting that stressful events cause a greater increase in the arousal network (e.g., locus coeruleus(LC) and periaqueductal gray) of women than men following exposure to aversive (threat) stimuli [13]”: In this place Authors should decide: do they want to build the hypothesis of sex-specific predisposition base or hormones-specific predisposition.

I think I should emphasize my doubts more: The context is extremely important here. Thesis: the stress response is variable, and depends on the set and changes in hormone concentration over time, is a biological thesis. Thesis: women react differently to stress because they are women, it can be interpreted as sexist.

The subject should be linguistic, with care and nuances. The paper must be checked by a native speaker.

Author Response

Referee 3

The work is still being edited carelessly.

  1. No figure, the signature remains.
  2. Many linguistic mistakes. I don't feel qualified to judge about the English language and style, but, I notice only in the introduction:
  • A critical sex-specific predisposition to the occurrence of PTSD have been identified-> has been
  • Hypothalamic peptides and the HPA axis reciprocally control each other activities, moreover gonadal hormones could affect theirfunction. -> Hypothalamic peptides and the HPA axis reciprocally control each other activities, moreover, gonadal hormones could affect their function.
  • The interaction between sex hormones, the HPA axis and hypothalamic hormones….-> The interaction between sex hormones, the HPA axis, and hypothalamic hormones is complex and occurs on both genetic by controlling the expression of peptides via interaction with promo-tors of those peptides and epigenetic levels.
  • The interaction between sex hormones, the HPA axis and hypothalamic hormones is complex and occur on at both genetic by controlling the expression of peptides via interaction with promo-tors of those peptides and epigenetic levels -> The interaction between sex hormones, the HPA axis, and hypothalamic hormones are complex and occurs on both genetic by controlling the expression of peptides via interaction with promo-tors of those peptides and epigenetic levels.

Moreover:

„This hypothesis is endorsed by a meta-analysis of functional magnetic resonance (fMRI) data suggesting that stressful events cause a greater increase in the arousal network (e.g., locus coeruleus(LC) and periaqueductal gray) of women than men following exposure to aversive (threat) stimuli [13]”: In this place Authors should decide: do they want to build the hypothesis of sex-specific predisposition base or hormones-specific predisposition.

I think I should emphasize my doubts more: The context is extremely important here. Thesis: the stress response is variable, and depends on the set and changes in hormone concentration over time, is a biological thesis. Thesis: women react differently to stress because they are women, it can be interpreted as sexist.

The subject should be linguistic, with care and nuances. The paper must be checked by a native speaker.

We thank the Reviewer for helpful and insightful comments. We are very sorry for so many editing errors, we tried to correct them all.

  1. The figure is incorporated in the text. Moreover, the Figure and Tables are attached in the Supplementary Materials.
  2. Thank you very much for indicating specific errors in the Introduction, we corrected it according to your helpful suggestions. Moreover, we checked again the Manuscript carefully for linguistic errors. The Manuscript was earlier checked by native speakers in professional English Language Editing service American Journal Experts. The certificate of AJE is attached in the Supplementary Materials.  
  1. The Reviewer of course is right that the proper hypothesis is that women are at greater risk of PTSD developing due to changes in hormone concentration of the time. We corrected the part of the text: „This hypothesis is endorsed by a meta-analysis of functional magnetic resonance (fMRI) data suggesting that stressful events cause a greater increase in the arousal network (e.g., locus coeruleus(LC) and periaqueductal gray) of women than men following exposure to aversive (threat) stimuli [13]”:
  2.  We are very sorry for so many linguistic and editing mistakes in the Manuscript. We read the Manuscript carefully and corrected mistakes. The paper has been checked by native speakers in professional English Language Editing service American Journal Experts. The certificate of AJE is attached in the Supplementary Materials.